# Current Development of Minimally Invasive Repair of Pectus Excavatum (MIRPE)

**DOI:** 10.3390/children9040478

**Published:** 2022-03-31

**Authors:** Frank-Martin Haecker, Thomas Franz Krebs, Kai-Uwe Kleitsch

**Affiliations:** 1Department of Pediatric Surgery, Children’s Hospital of Eastern Switzerland, CH-9006 St. Gallen, Switzerland; thomas.krebs@kispisg.ch (T.F.K.); kai-uwe.kleitsch@kispisg.ch (K.-U.K.); 2Faculty of Medicine, University of Basel, CH-4056 Basel, Switzerland; 3Department of General, Visceral, Thoracic, Transplant and Pediatric Surgery, UKSH University Hospital of Schleswig-Holstein Kiel Campus, Arnold-Heller-Strasse 3, 24105 Kiel, Germany

**Keywords:** pectus excavatum, MIRPE, thoracoscopy, sternal elevation, cross bar-technique

## Abstract

For decades, open surgical repair was the only available method to treat congenital and acquired chest wall deformities (CWDs). In 1998, D. Nuss described a minimally invasive procedure for surgical repair of Pectus excavatum (PE). Today, the Nuss procedure is performed with increasing frequency worldwide and considered as the “gold standard”. After its introduction, the method experienced numerous modifications such as routine thoracoscopy and/or sternal elevation, increasing safety of the procedure. Placement of multiple bars and/or the so called cross-bar technique were introduced to correct complex CWDs. Standardized pain management, the introduction of cryo-analgesia and a standardized postoperative physiotherapy program including deep breathing exercises facilitate the establishment of an enhanced recovery after surgery (ERAS) process. However, the widespread use of the minimally invasive repair of pectus excavatum (MIRPE) procedure has been associated with a significant number of serious complications. Furthermore, several studies report near-fatal complications, not only during bar placement, but also during bar removal. This review focuses upon the most relevant modifications, including recent published surgical techniques of MIRPE, in order to describe current developments in the field.

## 1. Introduction

Pectus Excavatum (PE) represents the most common chest wall deformity (CWD), occurring in approximately one in every 300–400 births, and showing a male predominance (approx. 4:1 ratio) [1,2]. For decades, open surgical repair such as the Ravitch technique and its modifications [3,4,5,6,7] was the most commonly used method to operate on patients suffering from congenital and acquired CWDs. In 1987, Donald Nuss developed a minimally invasive technique of pectus excavatum repair (MIRPE) that required no cartilage or sternal resection to avoid several operative features of the open repair. After publication in 1998 [8], MIRPE was well established and has been performed with increasing frequency worldwide within the last 20 years. At the same time, the number of articles reporting on different aspects of PE treatment went up from approximately 300 (1980 to 1989) to more than 600 (2000 to 2009), confirming raised awareness of this topic. Within the last decade, we noticed another significant increase, to more than 1000 published papers (2012 to 2021). Today, MIRPE represents the worldwide used “gold-standard” for surgical repair of PE in adolescent PE patients as well as in adult patients, and even in children. However, despite being “minimally invasive”, not only minor but, in particular, major complications associated with the MIRPE as well as an increasing number of near fatal complications have been reported in the meantime and summarized elsewhere [9,10,11]. As a consequence, the method has experienced numerous modifications over the last 20 years. Modifications include patient selection and indication, patient positioning, skin incisions, bar passage (intra- vs. extra-pleural placement), shape/size or number of bars and bar fixation [12]. Notably, two substantial changes aimed primarily at preventing intraoperative cardiac injuries were introduced: routine unilateral and/or bilateral thoracoscopy, and routine sternal elevation. Furthermore, placement of multiple bars and/or the so called cross-bar technique were introduced, in particular to correct complex CWDs and to reduce risk of secondary bar displacement.

As mentioned above, a very large number of articles describing technical aspects, modifications, complications of MIRPE and other aspects of diagnostic work-up and PE treatment have been published since the original publication in 1998. Even today, there is ongoing controversy regarding effects of MIRPE on cardiopulmonary function, exercise tolerance, body posture, etc. Of course, efficient pain management is another substantial part of a successful treatment protocol and has also experienced contemporaneous modifications. Analgesic options include regional analgesia such as epidurals, paravertebral nerve blocks or catheters, intercostal nerve blocks or catheters, cryoablation or subcutaneous infusion catheters, and patient-controlled analgesia (PCA). Furthermore, oral pain medication and a standardized postoperative physiotherapy program including deep breathing exercises complete the comprehensive standard operating procedure and may facilitate the establishment of an enhanced recovery after surgery (ERAS) program [13].

This review is not entitled to take the complete literature into account, but focuses upon the most relevant modifications including a selection of recently published surgical techniques for MIRPE to summarize current developments in the field.

## 2. Patient Selection

Besides clinical examination, standard assessment of PE patients with a clinically severe deformity include thoracal computed tomography (CT) scan or magnetic resonance imaging (MRI), echocardiogram and electrocardiogram, pulmonary function test and psychological assessment if needed. Surgery is indicated if the PE patient presents with a severe pectus deformity and fulfills two or more of the following criteria: CT index > 3.25, significant progression of the deformity, significant deformity related clinical symptoms, cardiac or pulmonary compression, restrictive lung disease and significant body image disturbance [1,14].

In the initial publication by Nuss, the median age at surgery was 5 years, and no patient over the age of 15 years was included [8]. We have noticed that significant worsening of the CWD is observed during pubertal growth spurt and remaining flexibility and elasticity of the chest wall represent important variables for an excellent outcome. Within the community of pectus surgeons, there is a broad agreement today that the best age to obtain MIRPE is between 13 to 16 years old when the chest cartilage and bones are still malleable and easier to manipulate and bend into the correct shape. As a consequence, median age for repair has shifted to 15 years [12,14,15]. In contrast, some centers are reporting excellent results in repair of younger patients, advocating for repair after 3 years of age [16]. Vice versa, large series have described good results for pectus repair in adult patients [17,18,19,20]. Even if these studies demonstrate feasibility of MIRPE at any age and there is no general limitation to perform MIRPE, we consider a soft, elastic, flexible and pliable chest wall, as in young adolescents, as a basic precondition for an excellent outcome. In contrast, PE patients with a severe combined defect (Figure 1) and/or a rigid chest wall may need additional measures such as sternal osteotomy, rib cartilage resection, multiple bar placement, and even, perhaps, a combined open and closed procedure (Figure 2).

In 2016, Nuss et al. updated and summarized the step-by-step details of surgical technique and their own modifications including informative illustrations [21]. Other modifications such as patient positioning, skin incisions, bar passage (intra- vs. extra-pleural placement), bar fixation, and shape and size of bars are summarized in detail elsewhere [12].

## 3. Allergy Testing

Approximately 10 years after the initial publication, a preliminary report raised awareness of metal allergy caused by the implant [22]. Because the consequences of metal allergy may include early replacement or even removal of the pectus bar, exclusion of metal allergy before MIRPE is mandatory. Metal allergy, specifically to nickel, which is part of the standard stainless-steel bar, is the most common contact allergy in the United States and Europe. The use of nickel jewellery is very popular in these areas and is considered to be the main cause of allergen exposure [22]. Since the incidence of metal allergy to pectus bar implants is estimated to occur in about only 5% of PE patients, allergy testing is usually not part of routine preoperative assessment. However, positive patient or family history of metal allergy, metal sensitization and female gender were identified as indicating increased risk [23,24]. Applying a stricter screening process for metal allergies, Shah et al. discovered a much higher allergy rate than identified before [23]. However, some patients may test negative for a metal allergy, though subsequently develop an allergic reaction, thus emphasizing that not all tests have equal positive predictive values [25]. Furthermore, a recently published case series reported that pectus bar allergies occur not only with stainless steel bars, but also with titanium bars [26], which are recommended to be used in case of nickel allergy. Careful preoperative testing may reduce the overall incidence of stainless-steel allergies but may miss titanium bar allergies [26]. Surgeons performing MIRPE should be familiar with the presentation of metal allergy, which is not necessarily limited to a rash.

## 4. Thoracoscopy

The initial publication by Nuss described blunt retrosternal dissection with a Kelly clamp without thoracoscopy [8]. In 2002, the same group reported their experience using routine thoracoscopy to improve safety during mediastinal dissection [27]. The most severe MIRPE associated complication is cardiac lesion and/or perforation from insertion of the introducer, as well as from positioning of the pectus bar. Even if rare, cardiac perforation as inadvertent side effect during MIRPE remains a potentially lethal risk. The routine use of thoracoscopic guidance is helpful in avoiding major complications such us cardiac lesions and injury to the mammary vessels. Its introduction marked the first major modification and decreased the rate of cardiac lesions. However, it failed to eliminate cardiac perforations completely [9,10,11,28].

The true incidence of life-threatening complications and mortality is not known, as we do not know the overall number of procedures performed worldwide, and major adverse outcomes are underreported [10]. Even if cardiac injuries have fortunately remained rare, it is not possible to determine exactly how many cardiac injuries have occurred before or after the introduction of routine thoracoscopy. A survey of members of the chest wall international group (CWIG) reported an incidence of cardiac injuries during pectus bar placement with and without thoracoscopy as less than 0.05% [10]. However, specifics of these cases are based on incomplete data sources that pool a broad spectrum, including primary and secondary MIRPE procedures, cardiac injuries during pectus bar placement and removal, surgeons with different levels of expertise, etc. [10,15]. Furthermore, a recent analysis of contemporary practice in 50 ACS NSQIP-pediatric institutions (the American College of Surgeons 2012 National Surgical Quality Improvement Program-Pediatric) revealed that routine thoracoscopy is used in only 84% of the participating centers [29].

Thoracoscopy remains widely accepted as it affords an overall improved visibility during intra-thoracal dissection and an overall reported lower risk of other major complications such as lung injury. Even if today there is no evidenced based data available concerning the effect of thoracoscopy on the true incidence of cardiac injuries, we may notice that the majority of articles and studies reporting on near fatal complications were all published before 2011 [10,28,30,31,32,33,34]. To avoid these complications, we consider the routine use of thoracoscopy during MIRPE as mandatory.

## 5. Sternal Elevation

During MIRPE, meticulous retrosternal dissection under continuous thoracoscopic guidance is mandatory and a precondition for safe pectus bar placement across the mediastinum. This step is considered as a potentially dangerous maneuver. Visualization across the mediastinum is compromised in PE patients with severe defects. In particular in adult patients, stiffness and rigidity of the chest wall as well as the corresponding force required to elevate the sternum may hamper retrosternal dissection. As mentioned above, surgeons fear cardiac lesion and/or perforation as the most severe complication from insertion of the introducer as well as from positioning of the pectus bar. To prevent accidental injuries of mediastinal structures, various options were developed for sternal elevation technique (SET) prior to mediastinal dissection. The technique of forced sternal elevation during mediastinal dissection is notably used in severe adolescent and adult MIRPE cases. Expanding the retrosternal space with SET to avoid accidental lesions of mediastinal structures has been reported by numerous authors [35,36,37]. SET may facilitate safe intra-thoracal retrosternal dissection, bar passage, and as bar placement. Furthermore, SET may reduce the risk of intercostal muscle stripping, another well reported complication after MIRPE. As we notice less chest wall flexibility in adult patients, the bars require more force to rotate, which may result in intercostal muscle stripping [34].

In the recent literature, a range of authors report on the routine use of SET during MIRPE to improve the safety of the procedure. SET options include the “Crane” technique introduced by HJ Park [38], the use of the Rultract^®^ retractor with a specially designed parasternal retractor [39], an additional subxiphoid incision, and the use of various handhold devices such as the handheld sternal elevator, Langenbeck retractors or the Wolkmann bone hook. The intraoperative use of the vacuum bell for sternal elevation was established as routine procedure [15,21,34] (Figure 3). The safety of MIRPE has improved clearly as there was no further near-fatal cardiac perforation and/or fatal incident, such as bleeding from mammarian vessels, reported when a SET was applied intraoperatively [15,34]. In our opinion, SET during MIRPE should be a routine maneuver.

## 6. Cross Bar Technique

The initial publication by Nuss described the use of one pectus bar and one stabilizer [8]. With increasing frequency and widespread use, number of pectus bars as well as positioning have been modified [12]. In patients with a severe PE deformity such as Grand Canyon type, or in adult patients with a rigid and stiff chest wall, the use of a second bar, or in selected patients even a third or fourth bar, has to be considered [18,19,20,40,41]. Pilegaard reported in 2015 the use of two short bars crossed under the sternum [42], and one year later HJ Park published a case using the cross-bar technique for the correction of complex PE repair, with the advantage of avoiding depression on the lateral chest wall [43]. The same group analysed the use of parallel bars in comparison to the cross-bar technique and concluded that the cross-bar technique might be superior to the parallel bar insertion technique for correcting a wider range of CWDs, especially at the lower part of the depression [44]. On the first occasion, we applied the cross-bar technique in a 17-year-old male patient presenting with severe bilateral depression of the chest wall combined with obviously pronounced costal flaring with finally excellent outcome (Figure 4, Figure 5, Figure 6 and Figure 7). Based on the diagonal position of the bars, with the lateral ends resting on the lower rib cage, a potential secondary flaring using parallel bars is not only avoided but a costal flaring can actually be corrected, while the force of elevating the sternum is distributed on two bars, without compromising the lateral chest wall. Today, the cross-bar technique is an inherent part of our routine MIRPE program.

## 7. Pectus Bar Removal

Pectus bar removal (PBR) is the scheduled final step of MIRPE, usually performed approximately 3 years after insertion. PBR is often deemed as a “minor” procedure and scheduled in an outpatient setting. Opening the surgical incisions for PBR, the tip of the pectus bar when passing the substernal tunnel can usually not be visualized. Results of an online survey raised awareness about the risk of serious complications during PBR [45]. Even if PBR has a high safety profile, the procedure might be associated with major complications such as life threatening hemorrhage from various mediastinal structures. Previous surgery (open heart surgery, previous Ravitch procedure, or thoracotomy), vascular lesions (in particular mammary vessels) and lung injuries can be other sources of life-threatening complications [45]. In addition, patients who present with a wound infection and/or pericarditis after MIRPE should be considered at higher risk for life-threatening complications during PBR. Finally, serious and/or near fatal complications experienced at the time of primary surgical repair may increase the risk of complicated PBR [46]. To increase safety of PBR, some authors recommend using SET and/or thoracoscopy, not only for pectus repair, but also for PBR [47,48,49,50].

Prior to PBR, proper bar placement is considered as a crucial precondition and mandatory for safe PBR. In an effort to further decrease complications during PBR, bilateral opening of surgical incisions, unbending the bar and meticulous mobilization of the bar are recommended. To manage severe complications if they occur during PBR, the procedure should be performed in a hospital setting with adequate resources and personal including cardiac surgeons. If the postoperative course is uneventful, discharge on the same day is reasonable [46].

## 8. Serious Complications

The cardiovascular morbidity and mortality related to MIRPE appears more severe in comparison to open repair [51]. Even more than 20 years after its introduction, the real prevalence and type of life-threatening complications related to MIRPE and subsequent PBR are still unknown and underreported. As mentioned above, cardiac perforation from insertion of the introducer due to positioning of the pectus bar during PBR represents a rare but potentially lethal risk. Early case reports followed by comprehensive reviews [9,10,11,51] raised awareness of these risks, essential in ensuring optimal safety. Pectus surgeons have to be aware of these serious complications. As mentioned above, introduction of routine thoracoscopy as well as intraoperative SET has improved safety of MIRPE significantly and decreased the rate of MIRPE related serious complications, respectively.

## 9. Alternative Extra-Thoracic Approaches

To avoid surgery related intra-thoracal access, alternative surgical techniques have been developed. Harrison et al. developed the so called Magnetic Mini-Mover Procedure, in which magnets are used to pull the sternum up over a 2 year time frame [52,53,54].

The so called Pectus-up procedure uses a plate and a special designed screw for extra-thoracic implantation to pull up the sternum. Preliminary results are encouraging, and an increasing number of centers use the technique for surgical repair of PE [55,56].

## 10. Conservative Treatment of Pectus Excavatum

There is no doubt that MIRPE is less invasive in comparison to open pectus repair. However, conservative treatment of PE using the vacuum bell (VB) has to be considered as less invasive in comparison to MIRPE. Many CWD patients present with a mild to moderate degree of CWD, which does not immediately warrant surgery. However, in particular, pediatric and young adolescent patients may benefit from some type of non-surgical treatment. Pain associated with postoperative recovery, the risk of imperfect results and the fact that surgical repair has to be performed under general anesthesia are the major reasons why patients want to avoid surgery. Due to these facts, the introduction of conservative treatment methods to correct PE as well as pectus carinatum (PC) has made this alternative therapy a focus of interest for patients, their parents and surgeons.

Pectus surgeons should be aware that specific conservative and/or surgical treatment is not necessary in every patient with a CWD. However, great attention should be directed to careful clinical examination. Clinical follow-up with observation is recommended, especially in pediatric and adolescent patients. Dependent on patient’s age and severity of the CWD, non-surgical treatment might be the first step in therapy. Similar to surgical repair, we consider elasticity and flexibility of the ventral chest wall as an important precondition for good to excellent success of VB therapy [57]. In our own study group, the youngest PE patient started at the age of 2 years, the oldest PE at 61 years [58]. Patient’s compliance and/or the size of the ventral surface to apply the VB represent limiting factors in starting very early. More details of frequency and duration of daily application of VB, stepwise increase of negative pressure and duration of treatment have been published elsewhere [57,58,59,60]. According to the current literature and our own experience, conservative treatment methods are more often used in PC patients than in PE patients. Orthotic bracing represents an effective, preferred and most often recommended treatment modality in the majority of PC patients. It should be considered as the first choice of treatment. This may be corroborated by the fact that we may find in the current literature many more studies on bracing than on surgical repair, concerning the specific treatment of PC [2,57].

In PE patients also, conservative treatment will be applied much more often than surgical repair. Larger series confirmed that approximately 80% of PE patients applied for conservative treatment using the VB. In contrast, approximately 20% underwent surgical repair [2,58,61]. VB therapy appears to be established as an additional, complementary tool in specific treatment of PE patients. In particular pediatric and young adolescent patients may be treated successfully [2,58,60,62,63,64,65,66,67]. Patients ≤ 11 years, chest wall depth ≤ 1.5 cm, chest wall flexibility, and VB use over 12 consecutive months were identified as predictive variables for an excellent outcome [51].

Patients and parents may appreciate the surgeon offering all of the different treatment modalities.

## 11. Conclusions

Current developments in MIPRE should include the routine use of thoracoscopy as well as intraoperative SET to improve the safety of the procedure and to avoid near-fatal complications such as cardiac injury. The majority of pectus surgeons perform the procedure in adolescent PE patients with a median age of 15 years. Preoperative assessment should include allergy testing, in particular in PE patients with personal or family history of metal allergy.

Individualized surgical repair is mandatory in PE patients presenting with complex CWDs. In PE patients presenting with concomitant bilateral costal flaring and/or depression of the lower part of the chest wall, we recommend the cross-bar insertion technique for surgical repair in order to achieve excellent cosmetic results.

For safe PBR, bilateral opening of surgical incisions, unbending the bar and meticulous mobilization of the bar is recommended. To manage severe complications if they occur during PBR, the procedure should be performed in a hospital setting with adequate resources and personal, including cardiac surgeons.

Conservative treatment of CWDs is less invasive than MIRPE. We consider conservative VB therapy as a substantial part of our comprehensive CWD treatment program. VB therapy has been established as an additional useful tool in specific treatment of PE patients, applicable to a significant majority.

## Figures and Tables

**Figure 1 children-09-00478-f001:**
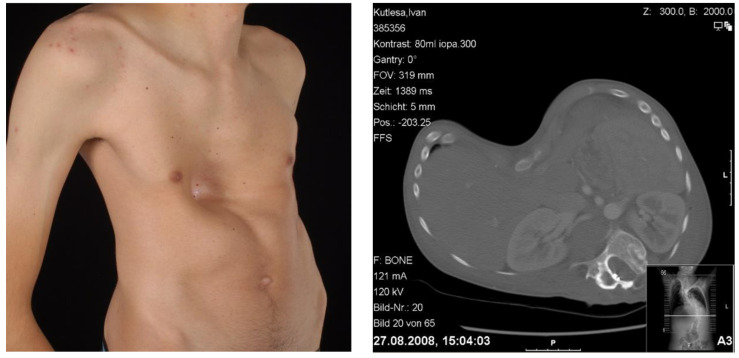
Clinical aspect and ct-scan of a 18 years old male patient with severe PE and combined severe scoliosis.

**Figure 2 children-09-00478-f002:**
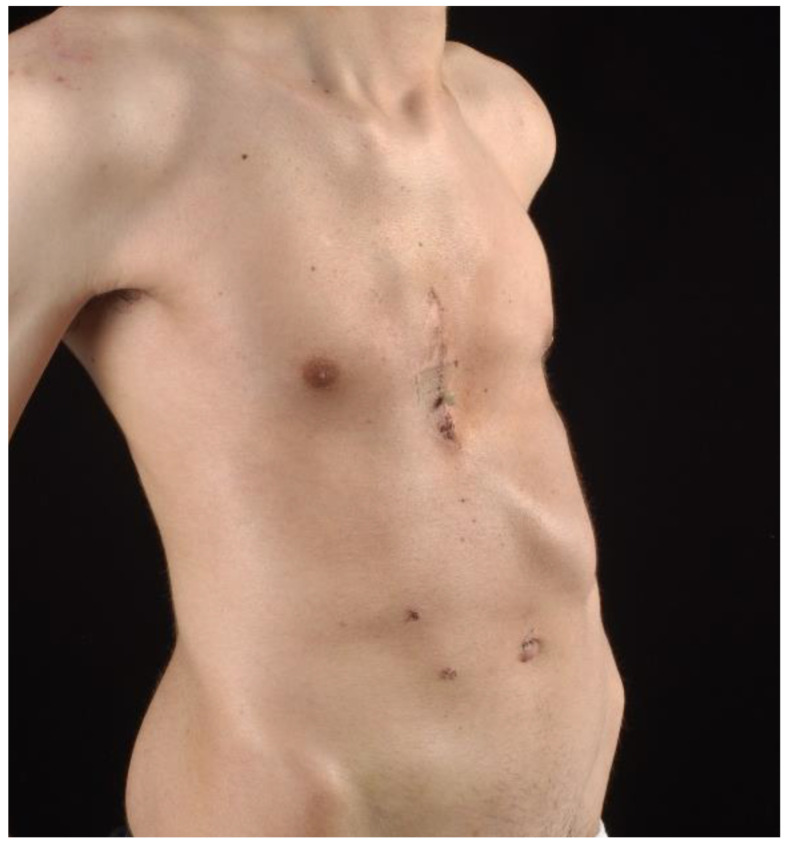
Same patient, clinical aspect after open pectus repair.

**Figure 3 children-09-00478-f003:**
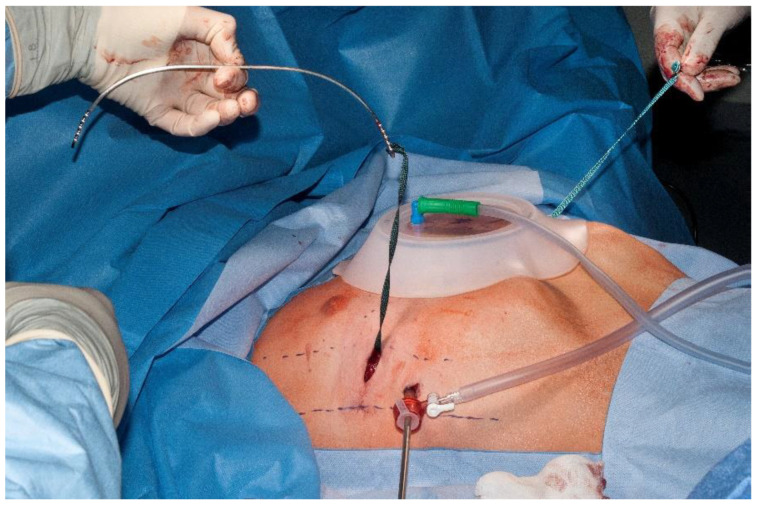
Intraoperative application of the vacuum bell.

**Figure 4 children-09-00478-f004:**
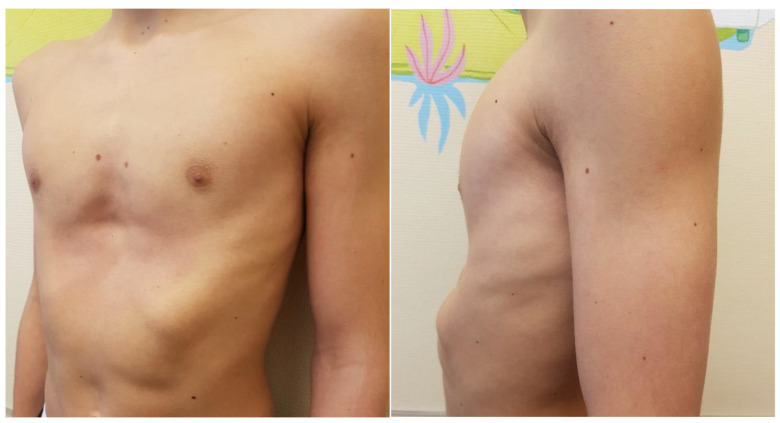
17 years old male patient with PE and bilateral costal flaring, before MIRPE (Cross bar technique).

**Figure 5 children-09-00478-f005:**
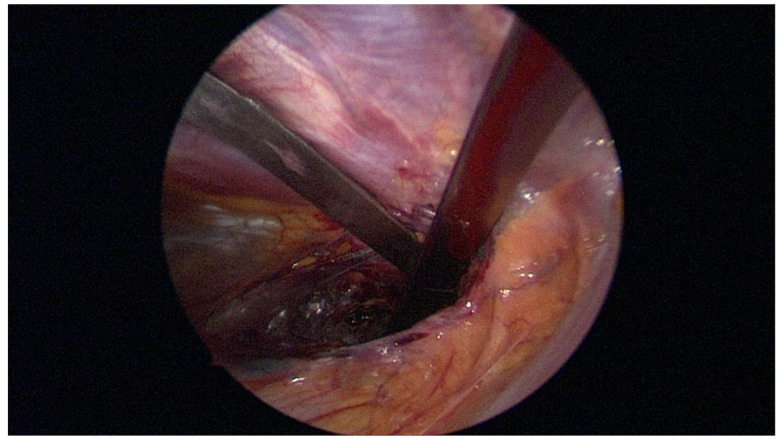
17 years old male patient with PE and bilateral costal flaring, thoracoscopic view (Cross bar technique).

**Figure 6 children-09-00478-f006:**
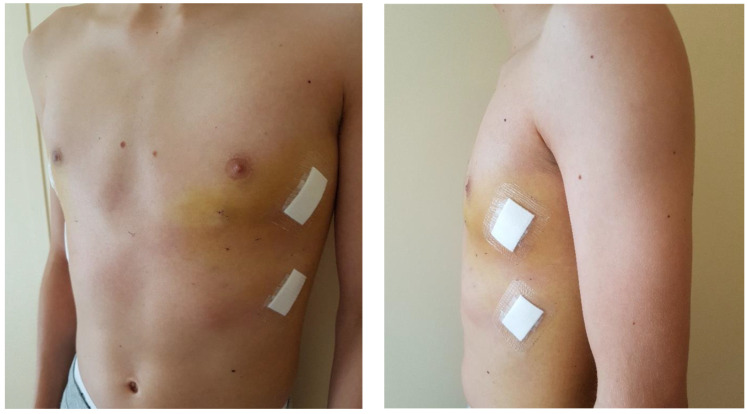
17 years old male patient with PE and bilateral costal flaring, after MIRPE (Cross bar technique); notice mild hematoma as side effect of intraoperative application of the vacuum bell.

**Figure 7 children-09-00478-f007:**
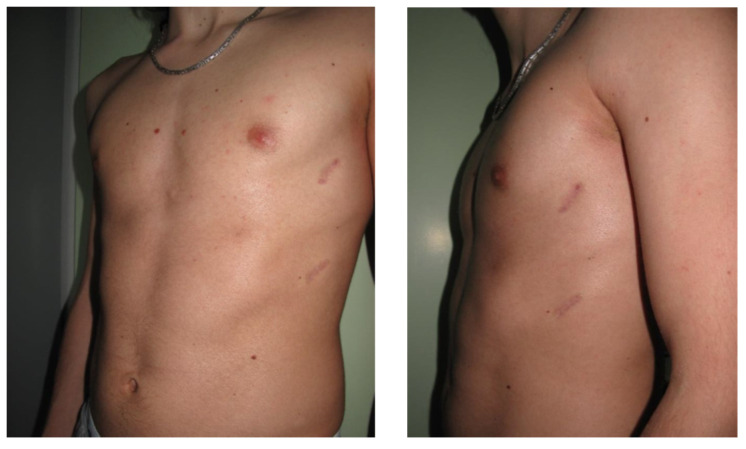
17 years old male patient with PE and bilateral costal flaring, 1 year after PBR.

## Data Availability

Not applicable.

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
