# Peer review of "Current Development of Minimally Invasive Repair of Pectus Excavatum (MIRPE)"

_children, 2022, doi:10.3390/children9040478_

Round 1

Reviewer 1 Report

The authors have significantly improved their manucript.

Two minor points which in the previous submission were supposedly addressed yet in the present manuscript I do not see added; a) removal of brand names from patient figures and b) intraoperative images.

Please add on the latest version

Author Response

Thank you for your valuable comments concerning our manuscript children-1650834. Please find our point-by-point reply below.

Reviewer 1:

  • The authors have significantly improved their manuscript.
  • Thank you for the kind re-assessment, appreciated.
  • Two minor points which in the previous submission were supposedly addressed yet in the present manuscript I do not see added; a) removal of brand names from patient figures and b) intraoperative images.
  • Removal of brand names: please check the latest submission of figures; we don’t see any brand names anymore.

intraoperative images: we added figure 3 and figure 5; furthermore, a reference displaying step-by-step details including informative illustrations was added (ref. 21); since we received the information that we may provide 2-3 figures, we considered the addition of 2 figures (fig. 3 and fig.5) and reference 21 as appropriate

Reviewer 2 Report

Congratulations on this new version of your manuscript. I would consider it for publication.

Author Response

Thank you for your valuable comment concerning our manuscript children-1650834. Please find our point-by-point reply below.

Reviewer 2:

  • Congratulations on this new version of your manuscript. I would consider it for publication. – Thank you so much!

This manuscript is a resubmission of an earlier submission. The following is a list of the peer review reports and author responses from that submission.

Round 1

Reviewer 1 Report

The authors present a narrative review of the literature on the minimally invasive technique of pectus excavatum repair. This is an excellent review focusing on patient selection, surgical technique and outcomes. It is well structured and written. The used references are up to date. 

Minor points

  • I would propose editing figures 1,2 in terms of any brands shown so as not to raise any copyright issues.
  • I would propose adding a separate section with limitations of MIRPE
  • Some intraoperative figures could make this review more illustrative for the readership

Author Response

Thank you for your valuable comment. Please find our reply below.

Reviewer 1:

  • I would propose editing figures 1,2 in terms of any brands shown so as not to raise any copyright issues.

Figures were edited, additional figures also do not display any brands.

  • I would propose adding a separate section with limitations of MIRPE

A paragraph and figures were added in the section “Patient selection”

  • Some intraoperative figures could make this review more illustrative for the readership

Intraoperative figures and a reference displaying step-by-step details including informative illustrations were added (ref. 21)

Reviewer 2 Report

Congratulations on this review manuscript. It is very interesting, although I would suggest some improvements or changes before considering it for publication.

I would suggest mentioning the technique "pectus up or Taulinoplasty". Here I add two cites about this procedure:

Bardají C, Cassou L. Taulinoplasty: the traction technique-a new extrathoracic repair for pectus excavatum. Ann Cardiothorac Surg. 2016 Sep;5(5):519-522. doi: 10.21037/acs.2016.09.07. PMID: 27747186; PMCID: PMC5056940.

Núñez García B, Álvarez García N, Aquino-Esperanza J, Esteva Miró C, Pérez-Gaspar M, Jiménez Gómez J, Betancourth Alvarenga JE, Santiago Martínez S, Jiménez-Arribas P, Güizzo JR. Efficacy and Safety of Taulinoplasty Compared with the Minimally Invasive Repair of Pectus Excavatum Approach to Correct Pectus Excavatum. J Laparoendosc Adv Surg Tech A. 2021 Dec;31(12):1402-1407. doi: 10.1089/lap.2021.0216. Epub 2021 Nov 29. PMID: 34847730.

About the pectus bar removal, have the authors found any article that include thoracoscopy during this removal?

Talking about Conservative Treatment of Pectus Excavatum, have the authors found any protocol that includes frequency of use of the vacuum bell, number of hours and minimum age?

Author Response

Thanky you for your valuable comment. Please find our reply below.

  • I would suggest mentioning the technique "pectus up or Taulinoplasty". Here I add two cites about this procedure 

    A separate paragraph entitled “Alternative extrathoracic approaches” including additional references was added.

  • About the pectus bar removal, have the authors found any article that include thoracoscopy during this removal? 

    Information including corresponding references (47-50) was added.

  • Talking about Conservative Treatment of Pectus Excavatum, have the authors found any protocol that includes frequency of use of the vacuum bell, number of hours and minimum age?

    The paragraph “Conservative Treatment of Pectus Excavatum” was completed with additional information about frequency, number of hours and minimum age. For more detailed information, corresponding references are listed.